**DOI: 10.1038/ncomms16024**　　**OPEN**

# Nonlinear cavity optomechanics with nanomechanical thermal fluctuations

Rick Leijssen[1], Giada R. La Gala[1], Lars Freisem[1], Juha T. Muhonen[1] & Ewold Verhagen[1]

Although the interaction between light and motion in cavity optomechanical systems is inherently nonlinear, experimental demonstrations to date have allowed a linearized description in all except highly driven cases. Here, we demonstrate a nanoscale opto-mechanical system in which the interaction between light and motion is so large (single-photon cooperativity $C_0 \approx 10^3$) that thermal motion induces optical frequency fluctuations larger than the intrinsic optical linewidth. The system thereby operates in a fully nonlinear regime, which pronouncedly impacts the optical response, displacement measurement and radiation pressure backaction. Specifically, we measure an apparent optical linewidth that is dominated by thermo-mechanically induced frequency fluctuations over a wide temperature range, and show that in this regime thermal displacement measurements cannot be described by conventional analytical models. We perform a proof-of-concept demonstration of exploiting the nonlinearity to conduct sensitive quadratic readout of nanomechanical displacement. Finally, we explore how backaction in this regime affects the mechanical fluctuation spectra.

[1] Centre for Nanophotonics, AMOLF, Science Park 104, 1098 XG Amsterdam, The Netherlands. Correspondence and requests for materials should be addressed to E.V. (email: verhagen@amolf.nl)

In cavity optomechanics, the interaction between light in an optical cavity and the motion of a mechanical resonator enables sensitive optical readout of displacement, as well as manipulation of the motion of the resonator through optical forces[1]. This has allowed demonstrations of sideband and feedback cooling of the mechanical resonator near its quantum ground state[2–5], squeezing of light[6–8] and of the mechanical zero-point fluctuations[9–11], entanglement[12] and state transfer[13] between the optical and mechanical degrees of freedom, as well as detection of radiation pressure shot noise[14,15] and non-classical correlations[16–19]. In all of these examples, the coupling between fluctuations of the optical field and the mechanical displacement can be regarded as linear for all intents and purposes.

However, the optomechanical interaction is inherently nonlinear. Indeed, the cavity optomechanical interaction Hamiltonian reads $\hat{H}_{int} = -\hbar g_0 \hat{a}^\dagger \hat{a} \hat{x}/x_{zpf}$, where $\hat{a}$ $(\hat{a}^\dagger)$ is the annihilation (creation) operator for the optical cavity field, $\hat{x}$ is the displacement operator for the mechanical resonator and $\hbar$ is the reduced Planck constant. The photon–phonon coupling rate $g_0 = (\partial \omega_c/\partial x)x_{zpf}$ quantifies the change of the cavity frequency $\omega_c$ due to a displacement the size of the zero-point fluctuations of the resonator $x_{zpf}$. This interaction Hamiltonian leads to nonlinear behaviour, as the equations of motion it generates contain products of two operators. The linearized form of the interaction $\hat{H}_{int} = -\hbar g_0 \bar{\alpha}(\delta \hat{a}^\dagger + \delta \hat{a})\hat{x}/x_{zpf}$ does not contain the nonlinear terms, but usually suffices to describe the dynamics of fluctuations[1]. This form emerges when the cavity field is written as the sum $\hat{a} = \bar{\alpha} + \delta \hat{a}$ of an average coherent field $\bar{\alpha}$ and fluctuations $\delta \hat{a}$, and the term containing $\delta \hat{a}^\dagger \delta \hat{a} \hat{x}$ is neglected by assuming $\delta \hat{a} \ll \bar{\alpha}$. The linearization is generally valid if the fluctuations $\delta \hat{a}$, insofar as they are induced by mechanical motion, do not approach or exceed the coherent field $\bar{\alpha}$. However, the assumption that $\delta \hat{a} \ll \bar{\alpha}$ is not valid if mechanical fluctuations shift the cavity completely in and out of resonance with the optical drive, that is, when they produce a cavity frequency shift comparable to the optical linewidth $\kappa$. Then, nonlinear processes become crucially important, and qualitatively different effects can occur.

In the quantum domain, intriguing implications of this nonlinearity are expected in the single-photon strong-coupling regime when the coupling rate $g_0$ exceeds the optical and mechanical loss rates $\kappa$ and $\Gamma$, respectively. There, quantum-level mechanical fluctuations induce a nonlinear response, creating non-classical states of both light and motion when the mechanical frequency $\Omega_m$ approaches the optical linewidth as well[20–22]. In the so-called bad-cavity limit ($\kappa > \Omega_m$), the nonlinearity of the interaction provides a useful path towards creating motional quantum states, for example through performing quadratic measurements of displacement (proportional to $\hat{x}^2$)[23–27].

In macroscopic or chip-based optomechanical implementations, the breakdown of linearity when $\delta \hat{a} \gtrsim \bar{\alpha}$ has so far only been experimentally relevant for mechanical resonators driven to large amplitude, for example through optomechanical parametric amplification. In that case, nonlinear effects determine the maximum amplitude of optomechanical self-oscillation[1,28–32] and can lead to complex nonlinear dynamical phenomena such as chaos[33–35].

Here, we establish and explore the regime where even intrinsic Brownian motion induces cavity frequency fluctuations larger than the optical linewidth. In this regime, the nonlinear nature of the cavity optomechanical interaction becomes important in all essential phenomena, including optomechanical displacement measurement and radiation pressure backaction. The regime is defined by $g_0\sqrt{2\bar{n}_{th}} \gtrsim \kappa$, where $\bar{n}_{th} = k_B T/\hbar\Omega_m$ is the average phonon occupancy of the mechanical mode with frequency $\Omega_m$, in thermal equilibrium at a temperature $T$ and $k_B$ is the Boltzmann constant. It is clear from this condition that any optomechanical system in which the ratio $g_0/\kappa$ is increased will enter this regime before reaching the single-photon strong-coupling regime, unless the mechanical resonator is pre-cooled to its ground state. The condition can equivalently be expressed as $C_0 \gtrsim \kappa/\gamma$, that is, the single-photon cooperativity $C_0 \equiv 4g_0^2/\kappa\Gamma$ being larger than the ratio of optical decay rate and mechanical thermal decoherence rate $\gamma \equiv \Gamma\bar{n}_{th}$. In our experiments, we follow a strategy of exploiting subwavelength optical confinement[36] to reach a single-photon cooperativity around $10^3$; two to three orders of magnitude larger than typical values in nanoscale optomechanical systems to date[1,37,38], and only comparable with cold-atom implementations[15,39]. In the systems demonstrated here, the apparent optical linewidth is dominated by the transduced thermal motion over a wide range of temperatures, and the transduction becomes extremely nonlinear. We numerically implement a model that describes transduction in this regime, in contrast to the conventional analytical description, which fails for fluctuations that approach the linewidth. Moreover, we analyse how the nonlinear response of the radiation pressure force to stochastic fluctuations alters the shape of mechanical fluctuation spectra. Finally, we provide a proof-of-concept demonstration of exploiting the nonlinearity to conduct sensitive quadratic readout of nanomechanical displacement.

## Results

**Sliced photonic crystal nanobeam.** Figure 1a shows the optomechanical system we employ. It combines low-mass, megahertz-frequency, nanomechanical modes with subwavelength optical field confinement in a sliced photonic crystal nanobeam[36], to establish strong optomechanical interactions with photon–phonon coupling rates $g_0$ in the range of tens of MHz. The fundamental mechanical resonance of the sliced nanobeam, shown in Fig. 1b, strongly influences the gap distance $d$ in the middle of the beam. The motion of the resonator in this mechanical mode is associated with a simulated effective mass of 1.5 pg, leading to relatively large zero-point fluctuations $x_{zpf} = \sqrt{\hbar/2m\Omega_m} = 43$ fm. As shown in the optical field profile of the fundamental cavity resonance of the structure in Fig. 1c, the nanoscale gap in the middle of the beam confines the light to a small area, which makes the optical cavity resonance frequency $\omega_c$ strongly dependent on the gap size $d$ between the two halves of the nanobeam. With the fabricated gap size of 45–50 nm, we simulated the optical frequency change due to a displacement of the beams to be $\partial\omega/\partial x/2\pi = 0.8$ THz nm$^{-1}$, where $x \equiv d/2$. This leads to an expected optomechanical coupling rate of $g_0/2\pi = 35$ MHz. We decrease the optical cavity decay rate and increase the outcoupling at normal incidence by engineering the angular radiation spectrum of the sliced nanobeam structure[40,41] (Methods section). The resultant simulated optical decay rate is 8.8 GHz, showing that the sliced nanobeam design is capable of combining large optomechanical interactions with relatively low optical losses ($Q > 10^4$). We believe that further optimization along these lines could, in principle, lead to still larger optical quality factors.

We employ a balanced homodyne detection scheme, schematically shown in Fig. 1d, to study the fluctuations imparted on the light in the nanobeam cavity through the optomechanical interaction (Methods section). Figure 2a depicts fluctuation spectra recorded with an electronic spectrum analyser, showing the two fundamental mechanical resonances of one device, measured at 3 K with the laser on-resonance with the cavity. We ascribe the two resonances to the two half-beams moving at slightly different natural frequencies, instead of the ideal antisymmetric eigenmode depicted in Fig. 1b. The fact that in this device the two resonances are nearly of equal strength indicates that the two half-beams are mechanically coupled at a

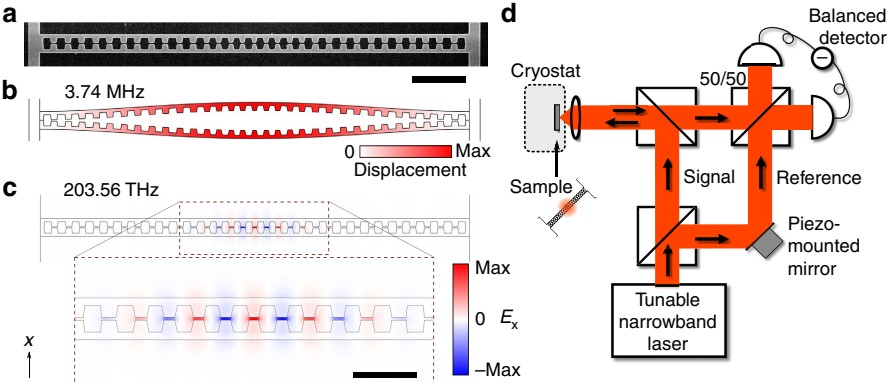

**Figure 1 | Structure and set-up.** (**a**) Electron microscope image of a silicon sliced nanobeam. The shown part is free-standing and has a thickness of 250 nm. The scale bar is 2 µm and is valid for **a**–**c**. (**b**) Simulated displacement profile of the fundamental mechanical resonance, which strongly modifies the gap size. (**c**) Simulated transverse electric field of the fundamental optical cavity resonance. The inset shows an enlarged view of the cavity region (scale bar 1 µm), formed by a tapered variation of the distances between, and sizes of, the holes. (**d**) Schematic diagram of the employed balanced homodyne interferometer measurement set-up. The reflection from the sliced nanobeam is interfered with the light from the reference arm, enabling near-quantum-limited measurement of fluctuation spectra even with low power incident on the sample (see Methods section for details).

rate smaller than their intrinsic frequency difference[36], and thus moving approximately independent of each other; stronger mechanical coupling would result in hybridization into symmetric and antisymmetric eigenmodes with different optical transduction strengths.

**Modification of optical response.** The signal strength of the transduced motion around the cavity resonance wavelength of 1457.5 nm, measured at 3 K and at room temperature, is shown in Fig. 2b as a function of laser detuning. The fluctuations are recorded while continuously sweeping the piezo-mounted mirror over multiple interferometer fringes, thus averaging the measured signal quadratures. As we analyse in detail elsewhere (La Gala *et al.*, manuscript in preparation), the resulting signal strength acquires a simple single-peaked detuning dependence with a maximum when the laser is tuned to the cavity resonance (Methods section). As Fig. 2b shows, the apparent linewidth of the optical resonance is strongly influenced by temperature. We infer from this that the frequency fluctuations of the cavity due to the thermal motion of the mechanical resonator dominate the response which occurs when they are larger than the intrinsic optical linewidth. This is illustrated in Fig. 2c: while the intrinsic optical response of the cavity is Lorentzian with linewidth $\kappa$ (orange thin lines), the distribution of cavity frequency fluctuations due to thermal motion has a Gaussian spectrum (brown thick line), whose linewidth $L_G$ is related to the root-mean-square (r.m.s.) value of the frequency fluctuations $\delta\omega_{r.m.s.}$ as $L_G = 2\sqrt{2\ln 2}\,\delta\omega_{r.m.s.}$. In the bad-cavity limit we consider here, we model the observed cavity response as a Voigt lineshape, which is a convolution between the Lorentzian cavity response and the Gaussian distribution of cavity resonance frequencies due to the Brownian motion. We note that the measured electronic power spectral density is proportional to the square of the optical response, which leads to a smaller apparent linewidth in the detuning dependence shown in Fig. 2b. In the following, we only report the extracted linewidth (Methods section), which directly corresponds to the optical loss rate $\kappa$ in the low-temperature limit, and the full width at half maximum of the frequency fluctuation distribution in the high-temperature limit.

As the thermomechanical displacement variance is given by $\langle x_{th}^2 \rangle = 2\bar{n}_{th} x_{zpf}^2$ (assuming dynamical backaction is negligible), the induced frequency fluctuations due to a single-mechanical mode

at frequency $\Omega_m$ are characterized by a r.m.s. amplitude

$$\delta\omega_{r.m.s.} \equiv \sqrt{\langle \delta\omega^2 \rangle} = g_0\sqrt{2\bar{n}_{th}} = g_0\sqrt{2k_B T/\hbar\Omega_m}, \quad (1)$$

which reveals a square-root dependence on temperature. In case multiple independent mechanical modes are coupled to the optical cavity, the variances of the cavity frequency fluctuations are added, that is, $\delta\omega_{r.m.s.}^2 = \sum_j \langle \delta\omega_j^2 \rangle$, which preserves the overall temperature dependence. Figure 2d shows the full measured temperature dependence of the apparent linewidth, which exhibits the expected square-root dependence on temperature at higher temperatures. We fit the data points using an equation that approximates the linewidth of the Voigt lineshape (Methods section), with a fixed Lorentzian contribution due to the intrinsic optical loss and a Gaussian contribution that follows equation (1). The resulting fit curve is shown in Fig. 2d, together with its asymptotes (thin blue lines). These asymptotes allow us to directly extract the intrinsic optical linewidth $\kappa$ and the variance of thermal-motion-induced frequency fluctuations of the cavity, without further calibration. There are two mechanical resonances that show significant coupling to the optical cavity resonance as shown in Fig. 2a, and we derive the ratio between their coupling strengths from the ratio between the signal strengths at the two mechanical resonance frequencies[36]. Using this ratio and the measured resonance frequencies, we obtain $g_0/2\pi = 24.7$ and 25.4 MHz for the two mechanical modes, which corresponds closely to the predicted value for an ideal antisymmetric mode of 35 MHz, since $\sqrt{24.7^2 + 25.4^2} = 35.4$.

This large optomechanical coupling rate, combined with the extracted optical decay rate of $\kappa/2\pi = 20.4$ GHz and the mechanical decay rate of $\Gamma/2\pi = 100$ Hz (measured at 3 K), means that this device has a single-photon cooperativity $C_0 = 4g_0^2/\kappa\Gamma = 1.1 \times 10^3$. The single-photon cooperativity is a metric that combines optomechanical coupling and losses[1,38]. It signals the inverse of the number of intracavity photons needed to perform a measurement at the standard quantum limit, if all photons escaping the cavity could be employed towards that measurement. Interestingly, the combination of quadrature-averaged detection with temperature-dependent linewidth measurement allows direct extraction of $C_0$, with no other calibration than that of the mechanical bath temperature. The extremely high value we report here, which exceeds previously reported nano-optomechanical architectures by two to three

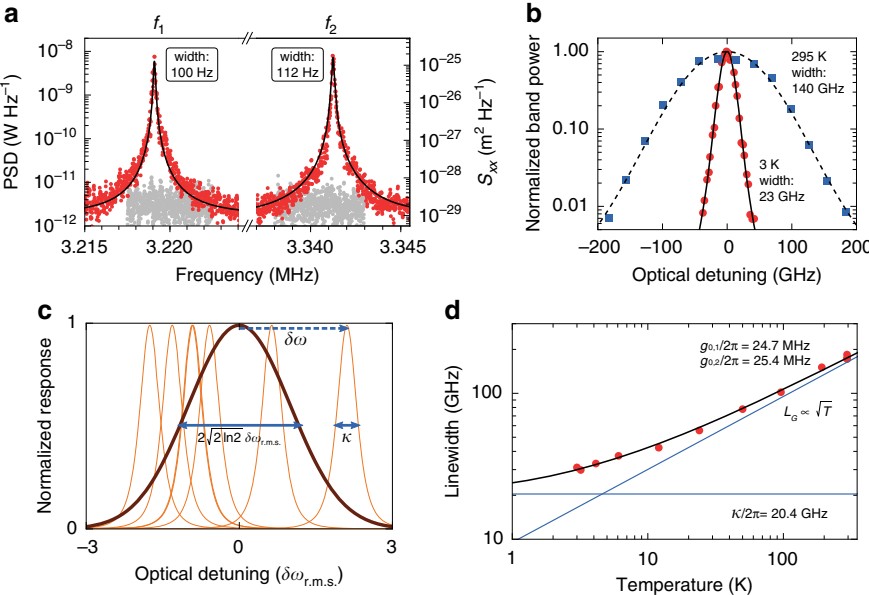

**Figure 2 | Optical linewidth broadening. (a)** Recorded optically measured spectra of the two fundamental mechanical resonances with frequencies $f_1$ and $f_2$ (PSD, power spectral density; optical power incident on sample: 11.3 nW). Grey noise spectra were recorded with the signal arm of the interferometer blocked. Black lines show the Lorentzian fit, used to determine the linewidths (full width at half maximum) shown in the figure. The displacement spectral density scale on the right-hand side assumes linear transduction of the known thermal motion of the structure at the cryostat temperature. **(b)** Detuning dependence of the measured transduced thermal motion, measured as the band power at $f_1$, the lowest mechanical frequency, at room temperature and at 3 K. The black solid and dashed lines show fits with a Voigt lineshape squared (Methods section), with the widths (full width at half maximum) shown. **(c)** Schematic representation of thermal-motion-induced linewidth broadening: the thin orange lines represent the intrinsic cavity response at a few example detunings, while the thick brown line shows the overall response resulting from averaging over the fluctuating detuning. **(d)** Optical linewidth versus temperature. The solid line is a fit with a model that assumes a constant Lorentzian intrinsic linewidth $\kappa$ convolved with a Gaussian with a width $L_G$ that depends on $\sqrt{T}$. The asymptotes (blue) of the fit function allow us to extract $\kappa$ and the optomechanical coupling rates $g_{0,i}$ for the two mechanical modes ($i = 1, 2$). Extracted values are shown in the figure.

orders of magnitude[1,37,38], highlights the prospects of such systems for measurement-based quantum control of motion.

**Nonlinear transduction.** When the relative cavity fluctuations $\delta\omega_{\mathrm{r.m.s.}}/\kappa$ are large, a typical mechanical oscillation samples the full width of the optical Lorentzian lineshape. Since this lineshape is nonlinear, higher-order harmonics are expected to appear in the transduced optical fluctuation spectrum. At room temperature, the cavity frequency fluctuations due to both mechanical modes have an amplitude $\delta\omega_{\mathrm{r.m.s.}} = (2\sum_j \overline{n}_{\mathrm{th},j} g_{0,j}^2)^{1/2} = 2\pi \times 69$ GHz $\approx 3.4\kappa$. As shown in Fig. 3a, at room temperature the measurement signal indeed contains fluctuations at (mixed) integer multiples of the two fundamental mechanical resonances, that is $f_{j,k} = |jf_1 \pm kf_2|$, where $j, k \in \{0, 1, 2, \ldots\}$. Around the fundamental frequencies near 3.3 MHz, we observe odd mixing terms up to ninth order (for example a peak is visible at $f_{5,-4} = 5f_1 - 4f_2$), and around the sum frequency at 6.6 MHz, even mixing terms up to tenth order can be identified.

At a temperature of 3 K, the ratio between the cavity frequency fluctuations, caused by both mechanical modes, and the intrinsic optical linewidth is $\delta\omega_{\mathrm{r.m.s.}}/\kappa = 0.34$, which still leads to significant higher-order transduction. Figure 3b shows a direct comparison of the spectra obtained at room temperature and at 3 K. As a measure for the higher-order transduction, we take the ratio between the second- and first-order transduction. The ratio between the orders is independent of other parameters, such that it gives direct insight in the strength of the nonlinearity. This ratio is clearly larger at higher temperature (its inverse is indicated with arrows in Fig. 3b). The difference in the signal-to-noise ratio between low and high temperatures is due to the mechanical linewidth being smaller at low temperatures (by a factor of 2) as

well as a redistribution of modulation power among higher harmonics, as explained in the following.

Higher-order transduction (for a single-mechanical mode) has previously been described with an analytical model that is based on a Taylor expansion of the measurement output around the average detuning[23,24,36]. For low-amplitude modulation, the higher-order terms in this expansion can be approximated as independent. Mathematically, this is based on an order-by-order approximation $(\cos\Omega_m t)^k \approx 2^{-(k+1)} \cos k\Omega_m t$. The resulting expression for the maximum signal power measured at the (multiple of the) resonance frequency $k\Omega$ is

$$\langle P^2 \rangle_{k\Omega} = 2A^2 k! \left( \frac{2\langle \delta\omega^2 \rangle}{\kappa^2} \right)^k, \tag{2}$$

where $A$ is a constant that depends on the optical power as well as the coupling efficiency to the cavity (for details, see Methods section and La Gala *et al.*, manuscript in preparation).

In Fig. 3c, the dashed line shows the ratio between the second- and first-order transduction that follows from this order-by-order approximation, $(2\delta\omega_{\mathrm{r.m.s.}}/\kappa)^2$. The data points labelled I and II, which represent room-temperature measurements reported in refs 24 and 36, respectively, are still well-explained by the order-by-order approximation. For the devices under study, $\delta\omega_{\mathrm{r.m.s.}}/\kappa$ is so large that this approach breaks down. This is shown by our temperature sweep data, where we take the fitted area $A_{f_j}$ under the peaks at the fundamental frequencies and at twice the frequency (labelled in Fig. 3a), and plot their ratio: $(A_{2f_1} + A_{f_1+f_2} + A_{2f_2})/(A_{f_1} + A_{f_2})$. This ratio reduces to $A_{2f}/A_f$ if the two frequencies $f_1$, $f_2$ coincide, which supports directly comparing it to the single-mode prediction of the order-by-order

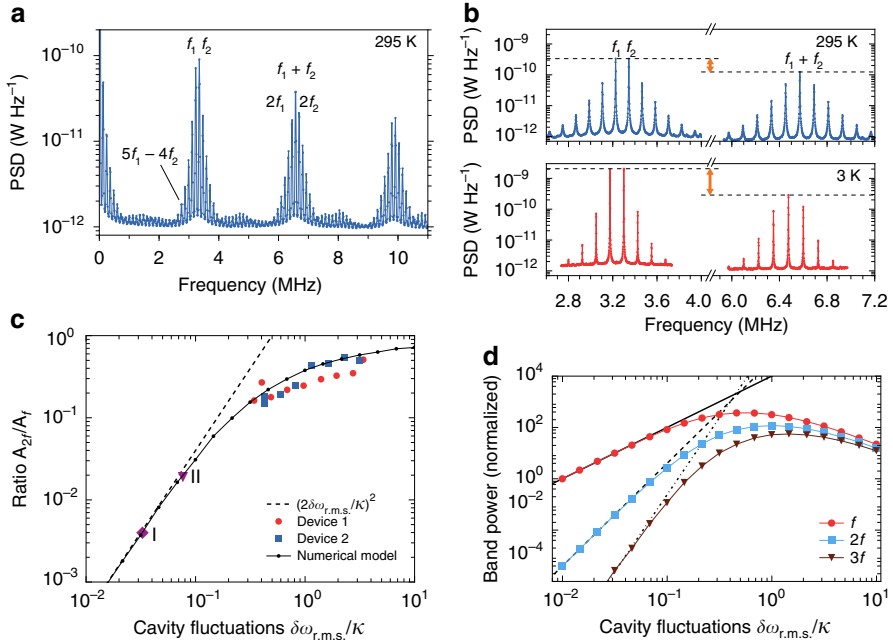

**Figure 3 | Nonlinear transduction. (a)** Power spectral density of transduced thermal motion, measured at room temperature (295 K). **(b)** Power spectral density of the group of peaks around the fundamental mechanical frequencies $f_1$, $f_2$ and around $f_1 + f_2$, at room temperature (blue) and at 3 K (red). The orange arrows indicate the ratio between the maximum first- and second-order transduction (dashed lines). **(c)** Ratio between the 3 second-order peaks around $f_1 + f_2$ and the peaks at the fundamental frequencies $f_1$, $f_2$ as a function of the relative cavity fluctuations $\delta\omega_{\text{r.m.s.}}/\kappa$. Device 1 is the same device presented in the other figures; device 2 is a similar device with different parameters from which more measurements are shown in Supplementary Note 2; room-temperature measurements from two recent publications are indicated with data points I (ref. 24) and II (ref. 36). The dashed line indicates the prediction from an analytical model with an order-by-order approximation, while the black data points connected with solid lines represent a calculation based on a numerically generated time trace to simulate thermal motion. **(d)** Expected band power at the fundamental frequency $f$ as well as at $2f$ and $3f$, as a function of the relative cavity fluctuations, normalized to the band power for $f$ at $\delta\omega_{\text{r.m.s.}}/\kappa = 10^{-2}$. Solid, dashed, and dash-dotted lines follow the order-by-order approximation, while the coloured points are the outcome of our numerical model.

approximation. The red circles indicate the results for the same device presented in the other figures, while the blue squares represent a different device with $g_0/2\pi = 10.8$ MHz and $\kappa/2\pi = 9.0$ GHz (Supplementary Note 2).

Where the order-by-order approximation breaks down, the nonlinear transduction still follows the prediction of our full model, in which we calculate the expected homodyne output from a numerical time-domain simulation of the thermal motion (Methods section). We then Fourier transform the simulated signal and plot the ratio of the band powers, noting that the numerical result does not depend on whether we include one or two mechanical resonances when plotted against total relative cavity fluctuations $\delta\omega_{\text{r.m.s.}}/\kappa$. This full model prediction is shown by the black dots connected by solid lines in Fig. 3c. The numerical simulation follows the experimentally observed trend, and shows that the ratio between second- and first-order transduction saturates close to unity at higher fluctuation amplitudes. We note that only such a numerical approach correctly takes into account the statistics of the transduced motion: the strongly nonlinear conversion of displacement to optical field precludes an analysis based solely on one or several moments of the displacement distribution. In other words, a single peak in the fluctuation spectrum at $k\Omega_{\text{m}}$ can no longer be dominantly associated with a specific scattering process involving $k$ phonons, but also contains contributions due to $k + 2, k + 4, \dots$ phonons. Although some of the data lie below the expected curve we note that this cannot be explained by poor thermalization of the sample. If the sample would not be thermalized at low temperatures, our estimate of $\kappa$ would be too high and our estimate of $\delta\omega_{\text{r.m.s.}}$ too low. Both such misestimations would lead to data points that lie above the predicted trend (black line) in

Fig. 3c, which is not what we observe. Independent measurements that vary optical power and radiation heat load also indicate good thermalization of the sample (Supplementary Notes 1 and 2).

Using the numerical model, we calculate the absolute signal power due to thermal motion at the fundamental frequency $f$ and its multiples $2f$ and $3f$ (points in Fig. 3d). For cavity frequency fluctuations $\delta\omega_{\text{r.m.s.}}/\kappa$ larger than about 10%, both linear and higher-order transduction no longer follow the order-by-order approximation given in equation (2) (black lines), which predicts monotonic increases in signal strength. Instead, we recognize an optimum single-harmonic signal strength due to cavity frequency fluctuations, beyond which larger fluctuations cause the cavity to be off-resonance most of the time. This reduces the transduction at a single harmonic in the optical signal, instead distributing energy equally among an increasing number of harmonics as the system operates deeper in the nonlinear regime.

**Quadratic measurement of motion.** A measurement that is directly sensitive to the square of displacement, $x^2$, can be used to estimate the energy of the mechanical resonator. In the resolved-sideband regime, displacement-squared optomechanical coupling hence provides a means to perform quantum non-demolition measurements of the phonon number[42–46]. But also in the bad-cavity limit, measurements of $x^2$ have been proposed as a possible route to preparing non-classical (superposition) states of motion of the mechanical resonator[23,24,47,48]. It is important then to suppress both linear measurement and backaction, either through structural design or through active feedback schemes[24]. Measurements of $x^2$ can be performed using the nonlinear transduction we study here, by detecting the transduced motion

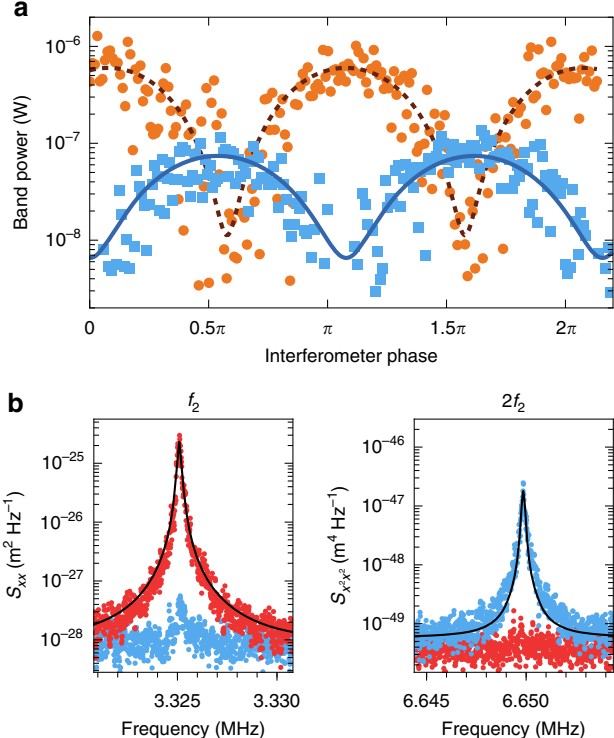

**Figure 4 | Quadratic measurement. (a)** Transduction of first ($f_2$) and second ($f_1 + f_2$) order peaks (orange and light blue data points, respectively) while sweeping the piezo mirror position, with 10.5 nW optical power incident on the sample. The solid lines show sinusoidal fits to the data. The horizontal scale was derived from the fit to the band power of $f_2$, and the data at $f_1 + f_2$ was shifted horizontally by about 3% to compensate for measurement drift. **(b)** Spectra at $f_2$ and $2f_2$ at optimal mirror positions for linear and quadratic transduction (red and light blue data points, respectively), taken with 12.8 nW of incident optical power. The black solid lines are Lorentzian fits, whose area was used to calibrate the vertical scale.

linewidth this level of imprecision corresponds to the ability to measure a phonon occupation of 752 with a signal-to-noise ratio of 1, which is an improvement over the measurement sensitivity in the proof-of-concept experiment reported in ref. 24 by approximately a factor 50. In the data shown here, the suppression of linear transduction is 28 dB, which could be further improved by implementing a feedback loop to lock the homodyne interferometer phase to a desired value.

It is important to note that our numerical model, as shown in Fig. 3d, predicts second-order transduction at only 10% of the strength of the order-by-order approximation for $\delta\omega_{r.m.s.}/\kappa \approx 0.34$, which is the size of the relative cavity fluctuations in our device at 3 K. Therefore, we expect that if this device is cooled further, to the point where the order-by-order approximation accurately predicts the transduction, an even lower imprecision noise of $6.3 \times 10^{-26}$ m$^2$ Hz$^{-1/2}$ would be recovered without any further optimization. This would be comparable to a phonon occupation of 238. In the regime where the order-by-order approximation holds, this imprecision, expressed as a phonon number $n_{imp}$ scales as

$$\frac{1}{n_{imp}} = 16 \left(\frac{g_0}{\kappa}\right)^2 \eta \sqrt{\frac{P_{in}\xi}{\hbar\omega_c\Gamma}}, \quad (3)$$

with $\eta$ the coupling efficiency to our cavity, $P_{in}$ the incident optical power, and $\xi$ a constant incorporating the technical homodyne detector measurement efficiency for both signal and local oscillator (Supplementary Note 3). With relatively modest improvements an imprecision at the single-phonon level could be within reach, for example by improving $g_0$ and $\kappa$ by factors of $\sqrt{2}$ and 2, respectively, as well as increasing $\eta$ to 40% from the 1.3% estimated in the current device (Supplementary Note 4), for example by employing waveguide-based coupling strategies[49]. The powers used for acquiring the data above are also low, corresponding to an average cavity occupation of $\sim 0.2$ photons. At higher powers, the mechanical spectrum was affected by low-frequency noise of the laser through the effects explored in the next section, an effect that could be ameliorated by for example shorter measurement times.

It is important to note that to use such measurements to create quantum superposition states, significant further advances are needed: first, the nonlinear measurement rate should exceed the thermal decoherence rate, requiring an imprecision $n_{imp} < (2\bar{n}_{th})^{-1/2}$. As such, lower bath temperature and larger intracavity photon number would be highly beneficial. Second, active feedback at the mechanical frequency would be needed to suppress quantum backaction associated with the existent linear coupling. As noted by Brawley et al.[24], suppressing this to the single-phonon level requires a measurement efficiency $\zeta$ that satisfies $g_0^2/\kappa^2 > (1-\zeta)/(8\zeta)$, where $\zeta = h\eta$, and $h$ is the quantum efficiency of our measurement set-up (Supplementary Notes 3 and 4). Fulfilling this thus requires a strong investment in reaching near-unity extraction efficiencies and larger optical quality factors in these devices.

at twice the fundamental frequency, $2f$, provided that the sample temperature is low enough such that the order-by-order approximation is valid. This gives rise to an effective quadratic coupling rate given by $g_0^2/\kappa$ (ref. 48), which amounts to $2\pi \times 32$ kHz in the device presented here. To put that number in perspective, one can compare it to the quadratic coupling rate $\mu_0 = (\partial^2\omega_c/\partial x^2)x_{zpf}^2$ in devices that are designed such that the frequency is directly proportional to the square of the displacement. A state-of-the-art double-slotted photonic crystal system recently demonstrated[45] a quadratic coupling rate of $\mu_0/2\pi = 245$ Hz.

In Fig. 4 we show selective linear and quadratic measurements of mechanical displacement at 3 K. Figure 4a shows the strength of the first- and second-order transduction (proportional to $x$ and $x^2$, respectively, in the order-by-order approximation) as a function of the piezo mirror position that controls the homodyne phase, which is now no longer continuously swept. The data follow the expected sinusoidal dependence.

We obtained the spectra in Fig. 4b by positioning the piezo mirror at the optimum points for first- and second-order transduction, depicted by the red and blue data points, respectively. The vertical scale was calibrated by using the order-by-order approximation, where the area under the peaks in the spectrum is proportional to the variance of the thermal motion of the structure, which here provides a lower limit for the sensitivity. This analysis yields an imprecision for the displacement-squared measurement of $2.0 \times 10^{-25}$ m$^2$ Hz$^{-1/2}$. To provide some context for this value, with the current mechanical

**Radiation pressure force with large cavity fluctuations.** The nonlinear regime not only impacts optical transduction of motion, but also pronouncedly affects the mechanical fluctuations themselves through its influence on radiation pressure backaction. The strongest manifestation of backaction in the regime where the cavity reacts nearly instantaneously to the mechanical motion ($\kappa \gg \Omega_m$) is the optical spring effect, which alters the mechanical resonance frequency depending on the detuning between a drive laser frequency and the cavity resonance[1]. Figure 5a,d show experimentally obtained spectrograms at low temperature (3 K, incident optical power

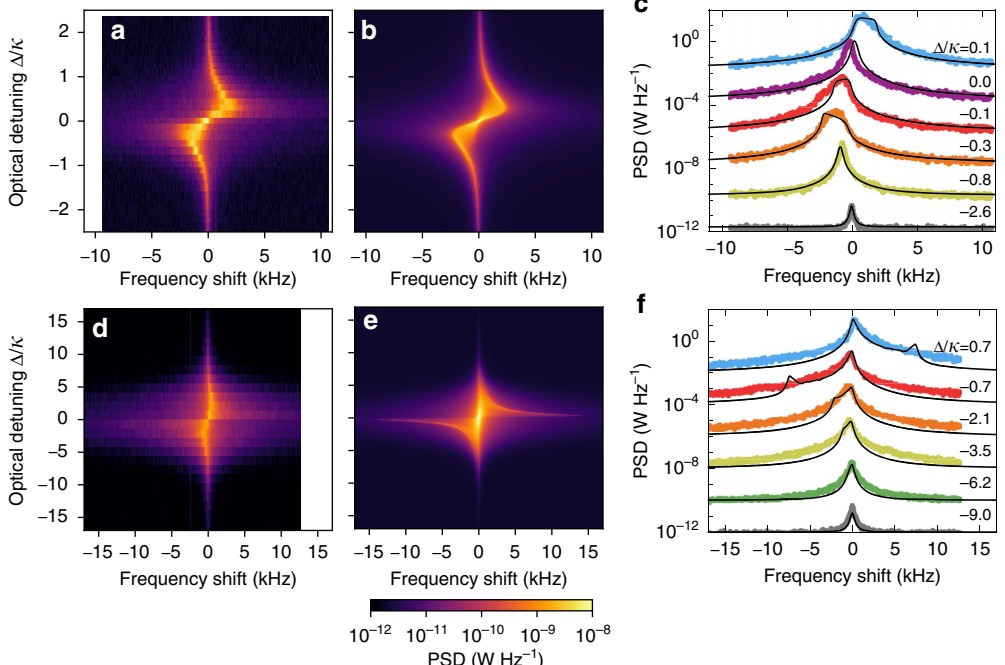

**Figure 5 | Optical spring effect modified by large cavity frequency fluctuations.** (**a,d**) Experimental spectrograms, showing transduced mechanical spectra (horizontal axis) versus the optical laser frequency (vertical axis) at two different experimental conditions: (**a**) cooled at 3 K, incident laser power 20.6 nW; (**d**) room temperature, 295 K, with incident laser power 124 nW. (**b,e**) Simulated spectrograms corresponding to those shown in (**a,d**), obtained by calculating frequency shifts for a large number of motional amplitudes sampled from a thermal distribution, and averaging Lorentzian lineshapes with centre frequencies given by the calculated shifts. The colour bar below panel (**e**) is shared for all the spectrograms (PSD, power spectral density). (**c,f**) Individual spectra at various detunings (coloured symbols), overlaid with the corresponding simulated spectrum (solid black lines). Spectra were offset by a factor of $10^2$ between them to avoid overlap.

20.6 nW) and at room temperature (295 K, incident optical power 124 nW) around the fundamental harmonic of one of the mechanical modes, while Fig. 5c,f show several cross sections at selected detunings. At low temperature, the observed effect is very similar to the normal (linearized) optical spring effect, where a blue-detuned laser shifts the mechanical frequency upwards and vice versa for red detuning. However, we additionally observe a broadening of the obtained spectra both when the laser is red- and blue-detuned by $\sim 10\,\mathrm{GHz}$ ($\sim \kappa/2$). At room temperature, the spectra also become asymmetric: instead of a symmetric Lorentzian lineshape, the peak in spectra close to the optical resonance has a much steeper edge on one side than on the other. In addition, the shift of the peak due to the optical spring effect usually scales linearly with the incident optical power, but the data set at room temperature shows a smaller, rather than a larger shift than the data set at 3 K, even though the used optical power is five times higher.

To account for these observations, we again need to take the large fluctuations of the cavity frequency due to thermal motion into consideration. The typical optomechanical model for the optical spring effect is based on the linearized equations of motion[1], which do not apply in the nonlinear regime we reach here. To model a single-mechanical mode, we instead calculate an effective spring constant from the first Fourier coefficient of the radiation pressure force, while the resonator oscillates harmonically, $x(t) = x_0 \cos \Omega t$. For this, we express the radiation pressure force in the cavity as

$$F_{\mathrm{rad}} = \frac{\hbar (\partial \omega_c / \partial x) n_c^{\max}}{1 + u^2}, \qquad (4)$$

where $u \equiv \frac{2}{\kappa} \left( \overline{\Delta} + (\partial \omega_c / \partial x) x \right)$ with $\overline{\Delta}$ the average detuning between the laser and cavity frequency, and $n_c^{\max}$ is the maximum

number of photons in the cavity when it is driven at resonance. The effective spring constant can be directly rescaled to obtain the frequency shift (Supplementary Note 5). As before, we numerically sample a thermal distribution to model the behaviour of the system, in this case by averaging over $10^4$ different amplitudes $x_0$, such that the displacement variance equals that from both mechanical modes in the experiment. Finally, we average the same number of Lorentzian lineshapes with the centre frequencies given by the simulated frequency shifts and their widths set to the median linewidth measured in the experiment, to approximate the intrinsic linewidth at that temperature (Supplementary Note 5). The resulting simulated spectra are shown in Fig. 5b,e and plotted as black solid lines in Fig. 5c,f. To allow overlaying the simulated spectra with the experimental cross sections, we rescaled all the simulated spectra such that the maximum value of the spectrum closest to resonance matched the experimental value. The model reproduces the broadening of the peaks as well as the asymmetry for the high-temperature data. At room temperature, a strong edge at larger frequency shifts is observed in the model, which is absent in the measurements. We attribute this to the presence of two mechanical modes, which will tend to soften this effect. An interesting feature in this model is that the spectra obtained with the laser on-resonance are not affected at all, and we indeed observed experimentally that at $\overline{\Delta}/\kappa = 0$ the effect is much smaller if not entirely absent.

We also observe that at higher optical powers, beyond $\sim 100\,\mathrm{nW}$ at 3 K, the influence of the nonlinear regime on the optical spring effect leads to imprinting of low-frequency noise on the mechanical motion as random drifting of the mechanical frequency. At higher powers still, self-oscillation regularly occurs. However, the fluctuations due to thermal motion can be larger than the amplitude at which a self-oscillating state would

normally saturate, leading to highly complex dynamics. We believe this leads to observed irregular behaviour, including random hopping of the oscillation frequency between the two mechanical modes, and occasional occurrence of instabilities for both blue and red detuning. Further study and modelling is needed to explore this regime.

## Discussion

Our analysis of both nonlinear transduction and backaction relies on a numerical model instead of an analytical description, as the typical order-by-order approximation to describe transduction breaks down at large relative cavity fluctuation strengths. This breakdown is related to the fact that the Taylor expansion used to describe the intracavity field[24] as a function of the relative detuning $u \equiv \frac{2}{\kappa}(\overline{\Delta} + \delta\omega)$ does not converge for all $u$. For example, at average detuning $\overline{\Delta} = 0$, the intracavity field can be written as

$$a = \frac{\sqrt{n_c^{\max}}}{1 + iu} \approx \sqrt{n_c^{\max}}(1 - iu - u^2 + iu^3 + u^4 \dots), \quad (5)$$

which does not converge for $|u| > 1$. At this detuning, $u = 2\delta\omega/\kappa$ represents the cavity frequency fluctuations due to the thermal motion, meaning the power series does not converge when the mechanical motion changes the cavity frequency by more than a linewidth. Therefore, this model cannot be used to extrapolate the expected sensitivity of optomechanical measurements in the nonlinear regime we describe here. Our numerical model relies on a direct calculation of the optical response to mechanical motion, which can be performed for any cavity frequency change $\delta\omega$. It is however crucial to correctly simulate the distribution of mechanical amplitudes within the thermal mechanical state.

We have demonstrated the effects of the optomechanical nonlinearity in a system that operates in the regime $\sqrt{2\overline{n}_{th}}g_0/\kappa \gtrsim 1$, and described how it modifies the optical response, transduction, and backaction over a wide range of temperatures. It is to be expected that a growing number of optomechanical systems will operate in this regime as parameters continue to improve, and the single-photon strong coupling regime ($g_0 > \kappa, \Gamma$) is approached. Indeed, various characteristics of the nonlinear regime we demonstrate are reminiscent of the expected effects due to quantum fluctuations in the single-photon strong coupling regime, including a modification of the optical response and the appearance of strong higher-order sidebands in optical fluctuation spectra. Whereas our sliced photonic crystal nanobeam devices demonstrate these effects for the bad-cavity limit, the regime has equally important impact for devices in the resolved-sideband regime ($\kappa \ll \Omega_m$), although precise manifestations are expected to vary. In particular, the optical excitation spectrum will be altered by displaying multiple discrete sidebands[20,21]. Moreover, backaction forces acquire an additional delay factor due to the longer lifetime of cavity excitation, which will lead to fluctuating damping and driving forces affecting the motion. As the manipulation of mechanical thermal noise is a topic of interest, for example, in sensing applications, it could be worthwhile to investigate the effects of the combination of dynamical backaction and the optomechanical nonlinearity. Notably, it would allow manipulating thermal fluctuations beyond a Gaussian distribution. Similar aims in the bad-cavity limit could be reached through nonlinear measurement, such as the displacement-squared measurements we demonstrated.

Interestingly, the alteration of the optical response due to thermal motion provides a new method to directly determine the intrinsic optical linewidth $\kappa$ and the optomechanical coupling rate $g_0$: the ratio of different harmonics of the transduced mechanical spectrum allows retrieving the relative frequency fluctuations $\delta\omega_{r.m.s.}/\kappa$. Together with a measurement of the

optical excitation linewidth (Methods section, equation (13)) this uniquely determines both $g_0$ and $\kappa$. This method only requires further knowledge of the bath temperature and the mechanical frequency. We note that there is no need for any other calibration, including characterization of optical powers.

The large single-photon cooperativity $C_0$ in the structures we present here offers prospects beyond the exploitation of the optomechanical nonlinearity, in particular for quantum measurement and control of mechanical motion. For example, the requirement for feedback cooling to the ground state (thermal occupancy below 1) is that the cooperativity $C_0 n_c > \overline{n}_{th}/(9\zeta - 1)$, where $n_c$ is the number of photons in the cavity[5,50,51]. In our sliced nanobeam structure, the measurement sensitivity is currently limited by the coupling efficiency, which we estimate to be $\eta \approx 1.3\%$ (Supplementary Note 4). The double-period modulation method we use to improve the coupling efficiency of a photonic crystal nanobeam cavity at normal incidence has previously been shown[41] to yield a simulated collection efficiency of more than 20%. Therefore we expect that by either improving the optical design, or by employing waveguide-based coupling schemes[49,51], the coupling can be further increased in our current free-space set-up. For example, with a detection efficiency of 2/9, ground state preparation would be in reach at a very modest minimum cavity occupation of $n_c > 17$ photons with the demonstrated parameters. Finally, the large optomechanical coupling strength in combination with low loss provides other opportunities for measurement-based control in the bad-cavity limit, such as conditional state preparation of the mechanical resonator by pulsed measurements[52], or quantum state swapping between the optical and mechanical degrees of freedom[27].

## Methods

**Sliced photonic crystal nanobeam design.** Eigenfrequency calculations were performed using finite element software (COMSOL Multiphysics). For optical simulations, perfectly matched layers were used to allow the extraction of the optical radiation losses. To estimate $\partial\omega_c/\partial x$, the optical frequency shift due to mechanical motion, $x$ (half of the gap size) was increased by 1 nm the full length of the nanobeam, leading to a change in the simulated eigenfrequency.

The sliced nanobeam design confines light in the transverse direction due to total internal reflection, while along the nanobeam the photonic crystal patterning creates a bandgap for light. An optical cavity is formed by a defect region in the middle with holes of different shape and periodicity, which is tapered to the periodic outer region over 5 holes to minimize optical losses[40]. In addition, we create a low-efficiency outcoupling grating in the structure by making the hole sizes alternatingly 5% wider and narrower. This double-period modulation allows part of the cavity field to scatter out at normal incidence[41], which is efficiently collected by our free-space optical measurement set-up. Our experimental results indicate that this strategy increases the coupling efficiency to the optical cavity from approximately 0.1% to 1–2% (Supplementary Note 4).

**Fabrication.** Devices were fabricated from a silicon-on-insulator substrate (SOITEC), with a 250 nm silicon device layer on top of 3 μm silicon oxide. Patterns were written using electron-beam lithography in an 80 nm layer of spincoated HSQ resist (FOX-15, Dow Corning) and developed using TMAH. The silicon layer was etched in an inductively coupled plasma etcher using a combination of $Cl_2$ and $HBr/O_2$ gases. Finally, the nanobeams were released to be free-standing by wet etching with HF, which also dissolves leftover resist and oxide-based deposits formed during plasma etching. The wet etch was followed by critical point drying to prevent collapse of the nanobeams. The device layer of silicon-on-insulator wafers typically contains compressive stress, which can induce buckling of the nanobeams even when using critical point drying. We avoided this by incorporating stress-relief features in the support structure around the free-standing nanobeams. We note that the fabricated devices exhibit optical quality factors that are somewhat lower (by a factor ∼2) compared to the designed value, likely due to remaining surface roughness, and structural deviation from the designed pattern.

**Measurement set-up.** A closed-cycle cryostat (Montana Cryostation C2) was used to control the sample temperature between 3 and 300 K. We used an aspheric lens positioned outside the cryostat window with an effective focal length of 8 mm a numerical aperture of 0.55 to focus the laser beam (New Focus Velocity 6725, linewidth ≤ 200 kHz) on the sample and to collect the reflection in free space. A balanced detector with two nominally identical photodiodes (New Focus 1817-FS)

detected the output of the homodyne interferometer, schematically shown in Fig. 1c. The detector signal was then Fourier transformed and the spectrum recorded with an electronic real-time spectrum analyser (Agilent MXA). The optical power in the reference arm was 135 μW or more, ensuring that the optical shot noise was at least as large as the electronic noise. The pressure in the cryostat was typically 0.3 mbar at room temperature, and well below $10^{-4}$ mbar at cryogenic temperatures.

**Sample thermalization.** A measurement of the power dependence of the transduced signal (Supplementary Note 1) was performed to verify that the measurement of the thermal motion of the nanobeam is not influenced by additional heating by the laser beam down to the lowest temperature. We then used the temperature sensor placed next to the sample in the cryostat to calibrate the scale for the displacement power spectral density $S_{xx}$ in Figs 2a and 4b. Measurements with different thermal radiation heat loads show that the presence of a cold window ensures that the sample is thermalized at all temperatures (Supplementary Note 2).

**Balanced homodyne interferometer signal.** The optical response of the balanced homodyne interferometer probing our cavity optomechanical system is a function of the average detuning $\bar{\Delta}$ between the laser frequency and the cavity frequency, the frequency shift of the cavity due to mechanical motion $\delta\omega$, the homodyne phase $\theta$, the cavity linewidth $\kappa$, and various constant factors such as the coupling efficiency to the cavity, and the optical power used for the measurement. To obtain the full expression, we start by describing the reflection from our nanobeam cavity using input-output theory

$$s_c = s_{in}\left(ce^{i\phi} - \frac{\eta\kappa}{i\Delta + \kappa/2}\right),\qquad(6)$$

with $\eta$ the coupling efficiency, $s_c$ is the amplitude of the reflected light and $s_{in}$ the amplitude of the input light. The term $ce^{i\phi}$ is due to non-resonant scattering from the nanobeam structure or the substrate. The total detuning $\Delta \equiv \bar{\Delta} + \delta\omega = \bar{\Delta} + \frac{\partial\omega}{\partial x}x$ is the sum of the laser detuning and the shift caused by mechanical motion. If $\phi = 0$, the resulting reflectivity shows a Lorentzian response, while other values will lead to the more general case of a Fano lineshape.

Using balanced homodyne detection, with the light in the reference arm, or local oscillator, described as $s_{LO} = |s_{LO}|e^{i\theta}$ and the input to the cavity $s_{in}$ taken to be real, the output of the detector is proportional to a virtual optical power $P$

$$\begin{aligned}\frac{P}{\hbar\omega} &= \left|\frac{i}{\sqrt{2}}s_c + \frac{1}{\sqrt{2}}s_{LO}\right|^2 - \left|\frac{i}{\sqrt{2}}s_{LO} + \frac{1}{\sqrt{2}}s_c\right|^2 \\ &= i(s_{LO}s_c^* - s_{LO}^*s_c) \\ &= |s_{LO}||s_{in}|\left(-2c\sin(\theta-\phi) + \frac{\eta\kappa(2\Delta\cos\theta + \kappa\sin\theta)}{\Delta^2 + \kappa^2/4}\right).\end{aligned}\qquad(7)$$

Substituting $\Delta \equiv \bar{\Delta} + \frac{\partial\omega}{\partial x}x$ yields the relationship between the measurement output $P$ and the displacement $x$.

**Optomechanical transduction with order-by-order approximation.** We consider the Taylor expansion of the measurement output $P$ for small fluctuations $\delta\omega$ around $\bar{\Delta}$

$$P(\bar{\Delta} + \delta\omega) = P(\bar{\Delta}) + \sum_{k=1}^{\infty}\frac{\delta\omega^k}{k!}\frac{\partial^k P}{\partial\omega^k}.\qquad(8)$$

For harmonic fluctuations $\delta\omega = \delta\omega_0\cos\Omega_m t$, the individual terms of this expansion contribute at different frequencies. To leading order, $\cos^k\varphi = \frac{1}{2^{k-1}}\cos k\varphi$, which means the higher-orders are completely spectrally separated, and we can consider this expansion order-by-order. If multiple mechanical modes are coupled to the optical cavity, the expansion will also include cross terms $\propto \cos(\Omega_1 t)^{j-k}\cos(\Omega_2 t)^k$, which are spectrally separated from higher-order terms in the same regime where the approximation holds for a single-mechanical mode. This means we can account for multiple modes by simply adding the variances they contribute in a specific order, and in the following we consider a single mode for simplicity.

We measure the power spectral density $S_{PP}(\Omega)$, whose integral over frequency gives the variance $\langle P^2\rangle$. Averaging over the homodyne phase, the detuning dependence for $(\partial^k P/\partial\omega^k)^2$ has a simple Lorentzian lineshape, raised to the power $(k+1)$, as we analyse elsewhere (La Gala et al., manuscript in preparation). The maximum contribution is therefore at resonance ($\bar{\Delta} = 0$) and can be expressed as

$$\left(\frac{\partial^k P}{\partial\omega^k}\right)^2 = A^2 k!^2\left(\frac{2}{\kappa}\right)^{2k},\qquad(9)$$

where $A^2 = 8P_{in}P_{LO}\eta^2$ is a constant prefactor that depends on the optical powers $P_{in}$, $P_{LO}$ in the signal and reference arm, respectively, and on the coupling efficiency to the cavity $\eta$.

We now calculate the band power at a frequency $k\Omega_m$, where only the $k$th term of the Taylor expansion contributes, as

$$\langle P^2\rangle_{k\Omega_m} \equiv \int\limits_{k\Omega_m} S_{PP}\,d\Omega = A^2\left(\frac{2}{\kappa}\right)^{2k}\left\langle(\delta\omega^k)^2\right\rangle_{k\Omega_m}\qquad(10)$$

Since $\delta\omega = (\partial\omega_c/\partial x)\delta x$, we can use the properties of the higher moments of $x$. For thermal motion, and again using the order-by-order approximation[24,53], we use

$$\left\langle(x^k)^2\right\rangle_{k\Omega_m} = \frac{k!}{2^{k-1}}\langle x^2\rangle^k,\qquad(11)$$

which can be directly substituted into equation (10) to obtain

$$\langle P^2\rangle_{k\Omega_m} = 2A^2 k!\left(\frac{2\langle\delta\omega^2\rangle}{\kappa^2}\right)^k,\qquad(12)$$

used in equation (2).

Supplementary Fig. 1 shows a spectrogram demonstrating the detuning dependence of the nonlinear transduction.

**Power spectral density and Voigt linewidth.** We plot the power spectral density of the electronic output signal of our measurement set-up, which has units of W Hz$^{-1}$. This corresponds to a power spectral density of a (virtual) optical power $P$, $S_{PP}$, which has units of W$^2$ Hz$^{-1}$. As a consequence, the Lorentzian detuning dependence of the optomechanical transduction leads to a Lorentzian-squared dependence in the power spectral density. Similarly, we observe the square of the Gaussian distribution of cavity frequencies due to thermal motion. Therefore, we fit the square of a Voigt lineshape, which models the convolution of a Lorentzian and a Gaussian lineshape, to our data, and extract the linewidth of the non-squared Voigt lineshape. This value then directly corresponds to either the optical loss rate $\kappa$ or the amplitude of the frequency fluctuations $\delta\omega_{r.m.s.}$, in the respective limits where $\kappa \ll \delta\omega_{rms}$ or vice versa.

We use an empirical equation for the linewidth of a Voigt lineshape[54]

$$0.5346\kappa + \sqrt{0.2166\kappa^2 + 8\ln 2\delta\omega_{r.m.s.}^2},\qquad(13)$$

where $\kappa$ is the linewidth of the Lorentzian lineshape and $2\sqrt{2\ln 2}\delta\omega_{r.m.s.}$ is the linewidth of the Gaussian lineshape. To fit the linewidth as a function of temperature, we substitute $\delta\omega_{r.m.s.} = \sqrt{2g_0^2 k_B T/\hbar\Omega_m}$. To account for multiple independent mechanical modes, we can calculate the individual optomechanical coupling rates from the asymptote of the fit curve if we know the ratio between the transduced peaks in the spectrum[36]. The variances due to the modes $j$ add up, $\delta\omega_{r.m.s.}^2 = \sum_j\langle\delta\omega_j^2\rangle$, which means equation (1) is modified to

$$\delta\omega_{r.m.s.}^2 = \sum_j\left(\frac{g_{0,j}^2}{\hbar\Omega_{m,j}}\right)2k_B T.\qquad(14)$$

**Numerical model for nonlinear transduction.** Time traces for the mechanical displacement $x$ were generated for one or two resonance frequencies $\Omega_m$. To simulate thermal motion, we randomly change the amplitude $A$ and phase $\varphi$ of harmonic motion $x = A\cos(\Omega_m t + \varphi)$. The points at which $A$ and $\varphi$ are changed are taken from a Poissonian distribution with mean time between jumps taken to be the mechanical damping time $\Gamma$. The new amplitude is taken from an exponential distribution characterized by a mean proportional to the average thermal occupation $\bar{n}_{th}$, while the phase is taken from a uniform distribution. A time trace of the measurement output $P$ was generated from the position time trace using our full model for the transduction (equation (7)) and phase-averaging the homodyne measurement. The discrete Fourier transform of this time trace allowed us to extract the signal strength at $\Omega_m$ and at higher-order multiples or mixing terms.

**Data availability.** All relevant data is available upon request.

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

## Acknowledgements

We thank Thijs Kleijntjens for assistance in setting up the balanced homodyne interferometer. This work is part of the research programme of the Netherlands Organisation for Scientific Research (NWO), and supported by the European Union's Horizon 2020 research and innovation programme under grant agreement No 732894 (FET Proactive HOT). E.V. gratefully acknowledges an NWO-Vidi grant for financial support. J.T.M. thankfully acknowledges funding from the European Union's Horizon 2020 research and innovation programme under the Marie Sklodowska-Curie grant agreement No 707364.

## Author contributions

R.L., J.T.M. and E.V. conceived the experiment. L.F. and R.L. fabricated the devices. G.R.L.G., L.F., J.T.M. and R.L. carried out numerical simulations, carried out experiments and analysed the results. R.L., J.T.M. and E.V. wrote the manuscript.

## Additional information

**Competing interests:** The authors declare no competing financial interests.

**Publisher's note**: 

