## [Peer Review File · Nature Communications]

Reviewers' comments:

Reviewer #1 (Remarks to the Author):

The authors present work on their split beam photonic crystal nanobeam, which has the largest single photon/single phonon coupling rate of any optomechanical resonator. Furthermore, even with relatively large optical loss rates, they show the largest single photon cooperativity ever reported. Since these authors are essentially working in uncharted parameter space, they must necessarily develop new analysis tools. This manuscript in part describes those new tools. One particularly intriguing new tool they show here is the ability to extract the single photon cooperativity from the optical linewidth, as long as they know the mechanical resonance temperature. This is quite a nice result. I am not fully convinced that the device is thermalized at all temperatures, and would like it if the authors could comment on the effect that poor thermalization would have. Why not use an additional phase modulation to ensure that you know your device temperature? In a related question, why does your signal to noise ratio actually go up when you decrease the temperature from 300K to 3K, as seen in figure 3b?

In addition to this new method of measuring the cooperativity in this extremely large optomechanical coupling regime, the authors show that they are able to control the degree of x^2 coupling in their measurement by controlling the phase of their homodyne interferometer. They produce an excellent result for their quadratic coupling rate and discuss the implications for Fock state measurement. In my opinion this is an important area for the future of optomechanics, and this manuscript adds significantly to the body of work in this area.

Overall, this is an excellent manuscript. It is well written, well analyzed, and is now the state-of-the-art in terms of optomechanical resonators. If the authors would include some extra discussion of the thermalization of their device, it should be immediately published in Nature Comm.

Reviewer #2 (Remarks to the Author):

This paper presents results from experiments on a free-standing photonic crystal nanobeam. The system is designed in such a way that it displays a very large single-photon optomechanical coupling rate between the confined optical mode in the beam and the motion of the beam. The coupling rate is large enough that the thermal motion of the beam shifts the frequency of the optical mode by more than its intrinsic linewidth at room temperature. At 3 K, the optical frequency shift is still a significant fraction of the linewidth. This is a parameter regime that has not been realized before and one where the standard linearized theory of cavity optomechanics breaks down. That being said, the system is still far away from realizing the even more interesting regime where quantum fluctuations lead to breakdown of linearized optomechanics. Nevertheless, it is worth noting that the single-photon cooperativity - an important figure of merit - is several orders of magnitude larger than in comparable experimental setups to date.

The results presented in the paper show how the large cavity frequency fluctuations induced by the thermal motion affect the systems' optical response to a laser drive. They also show how a measurement of mechanical displacement is affected, as well as the back-action associated with such a measurement. Finally, the ability to selectively measure the square of displacement in this regime is demonstrated. The experimental results are compared to numerical results from a nonlinear model, showing qualitative and, to a varying degree, quantitative agreement.

I find the science presented to be convincing and the paper to be well-written with appropriate citations of previous literature. The realization of the regime where cavity frequency fluctuations due to thermal motion is larger than the cavity linewidth is certainly of great interest to both

theorists and experimentalists within the field of optomechanics. Even if the experimental results presented are not particularly surprising, I believe that the paper can inspire the development of new theoretical ideas or improved experimental setups.

Before potential publication of the manuscript, the authors should address the issues listed below.

1. In the introduction, the condition for when linearization is valid is given as " $\Delta \alpha \ll \bar{\alpha}$ ". This claim is often found in the literature, but I question how precise this is. If the optomechanical coupling rate is very small, I would think that linearization is valid for all drive strengths, even very weak ones? I think the precise condition for the validity of linearization is rather that the cavity frequency fluctuations induced by the motion must be much smaller than the cavity linewidth, which the authors of course come to later in the introduction.

2. In Figure 1b, the "fundamental mechanical resonance" is depicted as a gap size fluctuation (a breathing mode). However, in the text, the authors talk of the individual resonances of the two half-beams, saying that they appear to be moving approximately independent of each other. This is a bit confusing. What is meant by "the fundamental mechanical resonance" in Fig. 1b?

In general, the discussion is sometimes as if there is just one relevant mechanical resonance (e.g. when describing the setup and when discussing Eq. (2)), and sometimes as if there are two independent resonances. I think the paper would benefit from clarifying this further.

3. Simulations give an optical decay rate of 8.8 GHz, which is referred to as "low optical losses". I understand that efforts have been made to decrease this parameter, but compared to other nanoscale system, it is not particularly small. Also, the measured value is 20.4 GHz - can the authors comment on the discrepancy between the simulated and measured values of both this parameter and those of the coupling rates?

4. When discussing Eq. (1), the displacement variance is related to temperature (page 6, line 161). Perhaps one should comment that this assumes no back-action?

5. In Figure 3c, there is some deviation between the results of the numerical model and the experimental results. This might deserve a comment.

6. The statements that a measurement of the displacement squared is equivalent to an energy measurement and that this can be used to perform a QND measurement of phonon number are conditioned on being in the resolved sideband regime (which is opposite of the regime studied in this experiment). This deserves a comment.

7. When discussing the quadratic measurement, it should be made clear at an early stage that it is only in the limit where the order-by-order approximation is valid that it is a measurement of displacement squared. For larger fluctuations, higher order terms will also contribute to the power at the frequency $2f$. Also, when a system is designed with no linear coupling (as in Ref. 46), no information about the displacement itself exits the cavity. Here, however, the information about linear displacement leaks out, even though it is not measured for a specific choice of homodyne phase. Can the authors comment on how this affects the usefulness of the quadratic measurement?

8. The description in page 12 of the numerical calculations shown in Figure 5 is a bit scarce. This should be expanded, either in the main text or in the Supplementary. It is for example not clear to me why the widths of the Lorentzians is set by the median linewidth measured.

9. When discussing Eq. (5), the crucial smallness parameter is not u , but rather $2\Delta\omega/\kappa$. In other words, I do not think the value of the detuning itself is relevant.

10. Regarding the discussion of the strong coupling regime on page 13, line 410: The optical excitation spectrum was analyzed also at nonzero temperature, both in Ref. 20 and Ref. 21.

11. On page 13, line 415, the possibility of using backaction to manipulate thermal fluctuations beyond a Gaussian distribution is mentioned. Citations would be in order here, as well as reasons for why the creation of non-Gaussian states is a goal in itself.

Reviewer #3 (Remarks to the Author):

The manuscript by Leijssen et al. describes a cavity optomechanical system with exceptionally large optomechanical coupling. They have developed a novel split photonic crystal design that provides a single photon cooperativity that is orders of magnitude larger than existing devices. In this system, thermally driven motion of the nanobeam modulates the resonance frequency of the photonic cavity by several times the cavity linewidth, putting the system into a highly nonlinear regime. This is the first such experiment to explore this qualitatively new regime of high order nonlinearity. Generally the main text of the paper is well organized, clearly written, and the data presented in the figures, for the most part, matches well with the numerical simulations. I believe that the demonstrations presented in this work will inspire thinking about new nonlinear optomechanical protocols. However, I do have one major concern and few minor questions that should be addressed:

1) I find the section "Quadratic measurement of motion" to be potentially misleading. Eq. (3) as defined appears to be a correct expression for the peak to shot-noise-background SNR in an integration time of $1/\Gamma$. However, the relevant signal to noise for single-phonon detection or non-classical conditional state preparation should be taken over an integration time of the quantum state lifetime $\sim 1/(\Gamma n_{th})$. For the same reason, the factor of n_{th} is (correctly) included in the inequality on page 5, line 431. For MHz scale mechanical frequencies n_{th} is a large factor even at cryogenic temperatures. Perhaps the authors could point out if there are any applications that become possible by satisfying Eq. (3) with $n_{min}=1$ (that could not be accomplished by just making a linear position measurement and squaring the results).

The Quadratic measurement section also sweeps under the rug the question of linear backaction, only briefly mentioned on line 273. The criteria for low linear measurement backaction as laid out in Ref [24], (as I understand it, Although I do not understand all of the subtle exceptions described in [24]), is that either $g_0 > \kappa$ or roughly $g_0 > \kappa \sqrt{1-\eta}$ with the addition of active feedback to suppress the backaction. The current work is still far (several orders of magnitude) from satisfying this criteria even if the detection efficiency is improved by amount suggested by the authors. As it is written now, the manuscript (around line 302) may be misread to imply that all that is necessary is to improve the suppression of linear transduction and not the suppression of linear backaction.

As these quantum state measurement and production ideas seem to be the main purpose of quadratic measurements, the authors should more carefully assess the suitability of their system

minor concerns:

2) Are there any indications of instabilities in this system, or are the light levels too low to induce self oscillations, bistability, photothermal or other parametric instabilities. What limits the light level to the 100nW scale? Also, are there any other noise sources that contribute to the cavity line broadening, higher order mechanical modes, low frequency acoustic noise, etc.

3) A few more details about the system would be useful. It is unclear what the 1.3% η includes. Is that just the spatial mode matching from the input round Gaussian mode to whatever the cavity emission pattern looks like. Is there internal loss in the cavity, or other losses in the homodyne detection system (photodetector quantum efficiency, additional mode matching losses, etc). All of these factors seem relevant for measurement based state preparation.

Response to Reviewer #1:

We thank the Reviewer for his/her careful reading of our manuscript. We are very pleased to hear that the Reviewer finds it "an excellent manuscript", describes our work as state of the art, and recognizes its significant addition to the future of optomechanics.

The Reviewer asks a valuable question about the thermalization of our device, in particular to comment on the effect that poor thermalization would have. We have carefully sought to verify experimentally that the sample is thermalized at all relevant temperatures. Indeed, this is especially important in nanophotonic devices such as ours as laser heating could cause significant local temperature rise of the nanocavity at cryogenic temperatures. Supplementary Figure S2 shows the signal strength as a function of laser power at the lowest cryostat temperature. Its linear dependence shows that any temperature rise due to laser heating is negligible (at both low and high temperature).

There could of course be other reasons why the sample temperature is different than that of the calibrated thermometer (which is mounted close to the sample on the same cold finger of the cryostat). However, at elevated temperatures we can assume the sample to be well thermalized. At those temperatures, the cavity linewidth is dominated by mechanical fluctuations as can be seen from the fact that the linewidth follows the expected \sqrt{T} law. We can thus conclude that our estimate of g_0 (derived from those measurements) is not affected by poor thermalization. At 3 K, however, we cannot exclude a priori that the sample temperature is raised, especially because it is exposed to the thermal radiation of a nearby window at ~ 50 K. Indeed, when we remove that window such that the sample is exposed to thermal radiation of the outer window at 300 K, we observe deviation of sample temperature by ~ 10 degrees. (This can be read off from Supplementary Fig. S5 in the revised SI as the horizontal difference between solid and dashed lines.) But the magnitude of this difference means that one would expect negligible temperature deviation with the ~ 50 K window in place: Assuming a T^4 dependence of heat transfer, the extrapolated temperature rise is ~ 0.1 K in that case.

This hypothesis is further supported by the fact that the relative heights of the transduced harmonics follow the expected theoretical trend (Fig. 3c). If the nanobeam would be badly thermalized at low temperatures, the actual value of κ would be smaller than our estimate and the value of $\delta\omega$ larger. This would lead us to underestimate $\delta\omega/\kappa$ and as a result the observed ratio of second to first harmonic plotted in Fig. 3c should then lie *above* the predicted curve (black line) around 300 K (rightmost blue and red points in Fig. 3c). At low temperatures (leftmost points), it should lie *even higher above* the predicted curve, as there our estimate of temperature would also be too low. This is not the case in Fig. 3c, with all data in fact *equal or lower* than the predicted curve.

Finally, the Reviewer gives the interesting suggestion to introduce a phase modulation to calibrate the frequency modulation. We had in fact considered this, but unfortunately such a technique is highly nontrivial in our case: The strong intrinsic cavity frequency modulation means that such a tone will inevitably mix with the mechanically-induced modulations, such that its magnitude is strongly altered and in fact depends on temperature itself. A second, practical, challenge lies in residual amplitude modulation, which is significant with respect to the cavity-transduced phase modulation in our devices as compared to higher- Q optomechanical cavities.

Action taken: We have followed the suggestion by the Reviewer to explain in the main text what would be the result of hypothetical poor thermalisation on page 9, line 270: "Although some of the data lies below the expected curve we note that this cannot be explained by

poor thermalisation of the sample. If the sample would not be thermalised at low temperatures, our estimate of κ would be too high and our estimate of the $\delta\omega$ too low. Both such misestimations would lead to datapoints that lie above the predicted trend (black line) in Fig. 3c, which is not what we observe. Independent measurements that vary optical power and radiation heat load also indicate good thermalization of the sample (Supplementary Notes 3 and 6).” We have also introduced additional discussion in the Methods (lines 565-568) and in Supplementary Note 6 of the estimation of possible temperature deviation in the presence of the cold window, and also comment on the intricacies of performing phase modulation temperature calibration at the end of Supplementary Note 6.

In his/her second (related) question, the Reviewer asks why the signal-to-noise ratio increases when we decrease temperature from 300 to 3 K in Fig. 3b. Let us first remark that we did not intend to directly compare the absolute scales of the top and bottom panels; instead, they serve to show the different relative heights of various harmonics. The two data sets were taken at slightly different optical powers, resolution bandwidths, and beam alignment. Nonetheless, the Reviewer is absolutely right in recognizing that the signal-to-noise ratio increases at lower temperature, even though the thermal occupation decreases. Two important factors contribute to this effect: First, the mechanical linewidth is smaller at low temperature by at least a factor 2. Second, due to the nonlinear transduction the area under the first harmonic peak (“Band power” in Fig. 3d) does not scale linearly with temperature as expected for traditional optomechanical systems: at high temperatures, the modulation power is distributed over many harmonics, strongly reducing the height of the first harmonic. This is shown as the deviation of the red points from the black line in Fig. 3d.

Action taken: We have explicitly mentioned the two mechanisms contributing to the increased signal-to-noise ratio at low temperatures in the revised manuscript (line 224).

We hope that we have clarified both questions, and that with the extra discussion that we included the Reviewer indeed finds our manuscript suitable for publication in Nature Communications.

Response to Reviewer #2:

We thank the Reviewer for his/her detailed assessment of our work and its merits. We are pleased to learn that the Reviewer finds the science we present convincing, of great interest to theorists and experimentalists, and that it can inspire the development of new ideas or setups.

The Reviewer asks us to address several issues before potential publication. We respond to each point of the Reviewer in the following and list the actions we have taken to improve our manuscript based on the Reviewer’s comments.

1. In the introduction, the condition for when linearization is valid is given as “ $\Delta a \ll \bar{\alpha}$ ”. This claim is often found in the literature, but I question how precise this is. If the optomechanical coupling rate is very small, I would think that linearization is valid for all drive strengths, even very weak ones? I think the precise condition for the validity of linearization is rather that the cavity frequency fluctuations induced by the motion must be much smaller than the cavity linewidth, which the authors of course come to later in the introduction.

We thank the Reviewer for bringing up this excellent point. Indeed, we have used the commonly used description for the condition for linearization. We agree with the Reviewer

that there is room for confusion in this general form. We believe that this confusion is resolved by a simple consideration, i.e., that the optical modulations δa find their origin in the mechanical motion. It is this assumption, that is commonly (often implicitly) made, that resolves the apparent incongruity: If modulations δa are caused by mechanical modulation of a drive field (proportional to \bar{a}), $\delta a/\bar{a}$ does not depend on the drive amplitude. In other words, for very weak drives, as the Reviewer suggests, the modulations imprinted on that drive through the mechanical motion are even weaker still (i.e. they scale with \bar{a}), and the linear regime is maintained regardless of drive strength. For standard (dispersive) optomechanical coupling, the two definitions ($\delta a \ll \bar{a}$ and $\delta\omega \ll \kappa$) coincide. And in our case, δa becomes comparable to \bar{a} precisely because $\delta\omega \approx \kappa$.

Action taken: we have replaced the sentence “The linearization is generally valid in the presence of a strong optical drive” by “The linearization is generally valid if the fluctuations δa , insofar as they are induced by the mechanical motion, do not approach or exceed the coherent field \bar{a} ” (line 53).

2. In Figure 1b, the “fundamental mechanical resonance” is depicted as a gap size fluctuation (a breathing mode). However, in the text, the authors talk of the individual resonances of the two half-beams, saying that they appear to be moving approximately independent of each other. This is a bit confusing. What is meant by “the fundamental mechanical resonance” in Fig. 1b?

In general, the discussion is sometimes as if there is just one relevant mechanical resonance (e.g. when describing the setup and when discussing Eq. (2)), and sometimes as if there are two independent resonances. I think the paper would benefit from clarifying this further.

The Reviewer rightly points out that we were not always as clear as we could be in denoting when we consider even and odd eigenmodes of an ideal system, when we consider the actual eigenmodes in our sample (which are linear combinations of the even and odd modes), and when we consider only a single mode for the sake of simplicity and generality. In various instances we have quoted general equations for single modes. As the fluctuations of the two mechanical modes observed in our sample are uncorrelated, those equations are readily extended to our case by adding the two modes’ variances.

Action taken: We have clarified the relation between the two measured modes and the ideal antisymmetric mode in the text on page 4 (line 131) and explicitly compare the measured optomechanical coupling rates (g_0) to the one predicted for the antisymmetric mode on page 6 (line 186). Moreover, we explicitly mention on pages 8 (lines 228 and 248) and 13 (lines 400 and 411), as well as in Supplementary Note 2 (line 41), that we write the equation for a single mechanical mode, and how we apply them to the case of two modes.

3. Simulations give an optical decay rate of 8.8 GHz, which is referred to as “low optical losses”. I understand that efforts have been made to decrease this parameter, but compared to other nanoscale system, it is not particularly small.

We agree that many optomechanical systems exhibit still larger optical quality factors. For a nanoscale system, and especially one that has sharp structural features and such strong field confinement as these, reaching damping rates of the order of a few GHz is not trivial. In fact, before the described efforts we did not know if these novel designs could really host optical Q s of $\sim 10^4$. They are an important contributing factor to the record-high single-photon cooperativities we report. But we are happy to put this qualification in more appropriate context.

Action taken: We have rephrased text on line 122 to “with relatively low optical losses ($Q > 10^4$)”, followed by a new sentence that discusses the possibility to design still higher Q s.

Also, the measured value is 20.4 GHz - can the authors comment on the discrepancy between the simulated and measured values of both this parameter and those of the coupling rates?

The discrepancy of simulated and experimental optical linewidths are most likely due to fabrication imperfections (surface roughness, sidewall verticality, stress, deviations of hole dimensions from design). Similar factors could play a role in the fact that the external coupling rate is smaller than the designed value. While the value of g_0 can be affected by small deviations of slit width, rounding of sharp corners, or stress, the measured values for the two nearly independent modes actually match our prediction of $g_0/2\pi = 35$ MHz for one hybridized mode very well, since $(24.7 \text{ MHz})^2 + (25.4 \text{ MHz})^2 = (35.4 \text{ MHz})^2$ (as noted above, this comparison has been explicitly included in the manuscript on page 6). We hope to gain more insight in the precise reason for the discrepancies (and further improve our devices) in future studies.

Action taken: We have introduced a sentence (line 542) that discusses the deviation and its likely cause.

4. When discussing Eq. (1), the displacement variance is related to temperature (page 6, line 161). Perhaps one should comment that this assumes no back-action?

We thank the Reviewer for noting this point, he/she is absolutely right.

Action taken: We added a remark on line 165 of the revised manuscript that this assumes dynamical backaction is negligible.

5. In Figure 3c, there is some deviation between the results of the numerical model and the experimental results. This might deserve a comment.

The Reviewer correctly points out that there is some discrepancy (within a factor 2 at maximum) between some of the observed peak ratios and the theoretical prediction. We do note that the model we compare to has relatively few parameters, and that for low values of $\delta\omega/\kappa$ the data approaches the prediction as it should.

Action taken: We have added a remark in the revised manuscript that comments on the discrepancy (line 270). Moreover, upon the recommendation of Reviewer 1 we discuss that the deviation cannot be due to a potential poor thermalization (which would cause the opposite deviation than that observed).

6. The statements that a measurement of the displacement squared is equivalent to an energy measurement and that this can be used to perform a QND measurement of phonon number are conditioned on being in the resolved sideband regime (which is opposite of the regime studied in this experiment). This deserves a comment.

We agree with the Reviewer. Our references to energy measurements are on the one hand intended to give context about quadratic measurements, and on the other hand to provide a sense of scale for the obtained imprecision of our nonlinear measurements. Nevertheless, although it is true that resolved sideband regime is needed for phonon number measurements, in the bad-cavity limit quadratic measurements can also be used to produce non-classical (superposition) states (see further answers to point 7 and Reviewer 3), as outlined in Jacobs et al., Phys. Rev. Lett. 102, 057208 (2009) and Brawley et al., Nat. Commun. 7, 10988 (2016) (refs. 48 and 24 in revised manuscript). We now make this point clearer and we also comment more on the requirements to use these measurements for non-classical state creation (see below).

Action taken: We explicitly mention the difference between quadratic measurements in the resolved and unresolved sideband regime on lines 291 and 293, and added a reference to Jacobs et al. Moreover, we have revised our discussion on page 11 lines 338-354 to put the

quantitative comparisons in better context.

7. When discussing the quadratic measurement, it should be made clear at an early stage that it is only in the limit where the order-by-order approximation is valid that it is a measurement of displacement squared. For larger fluctuations, higher order terms will also contribute to the power at the frequency $2f$. Also, when a system is designed with no linear coupling (as in Ref. 46), no information about the displacement itself exits the cavity. Here, however, the information about linear displacement leaks out, even though it is not measured for a specific choice of homodyne phase. Can the authors comment on how this affects the usefulness of the quadratic measurement?

These are both valid points and valuable suggestions. Information about linear displacement leaks out because of the presence of linear coupling. This coupling is also responsible for linear measurement backaction noise that disturbs the resonator state. This is caused by the photon number fluctuations in the cavity. As explained in reference 24, the measured signal at the fundamental frequency carries a record of these fluctuations. If the measurement efficiency is large enough, feedback of this signal could be used to suppress linear backaction to below the single-phonon level. We now explicitly comment on the requirements for this feedback in the quadratic measurement section (this was also suggested by Reviewer 3).

Action taken: To address the first point of the Reviewer, we added a statement “..., provided that the sample temperature is low enough such that the order-by-order approximation is valid” early in the section (line 302), a point that is repeated later (line 340 and further). Regarding the presence of linear coupling, we added a discussion in the revised manuscript (lines 298 and 362-373) with requirements to prepare nonclassical states with these methods, as also suggested by Reviewer 3.

8. The description in page 12 of the numerical calculations shown in Figure 5 is a bit scarce. This should be expanded, either in the main text or in the Supplementary. It is for example not clear to me why the widths of the Lorentzians is set by the median linewidth measured.

To address the specific question of the Reviewer, the mechanical linewidth of the device changes with temperature (when cooled down to 3 K, it decreases by at least a factor of 2 from its value at room temperature). Therefore we used the experimentally measured linewidth as an input for the model. The median linewidth was chosen because it is much less affected by the smearing out effect than the average linewidth. Since for most measurements in a full sweep of detuning the broadening is negligible, the median is a good representation of the intrinsic linewidth, as shown in the new Supplementary Fig. S4 in the revised SI. More generally, we thank the reviewer for pointing out that the description of the numerical calculation was not yet completely clear.

Action taken: We have amended the description of the employed model in both the main text (page 13, lines 410 and 415 as well as in Supplementary Note 5 (lines 142-154) to contain more detail about the calculation. In particular, we added a sentence that describes the motivation for using the median linewidth.

9. When discussing Eq. (5), the crucial smallness parameter is not u , but rather $2\delta\omega/\kappa$. In other words, I do not think the value of the detuning itself is relevant.

At zero detuning, the two conditions are equivalent. For different detuning $\bar{\Delta}$, one can derive that the convergence criterion is $(\delta\omega^2 - \bar{\Delta}^2)/(\kappa/2)^2 < 1$.* So in principle, detuning does

* *Outline of derivation:* The intracavity field takes the form $A/(1 + iu)$. Inserting $u = (\bar{\Delta} + \delta\omega)/(\kappa/2)$ and taking a Taylor expansion yields a form $B[1 - i(2\delta\omega/\kappa)/(1 + i2\bar{\Delta}/\kappa) - \dots]$, with B a constant. This

play a role in the convergence of the expansion, but in slightly different fashion than suggested by Eq. (5) and the surrounding original manuscript text.

Action taken: We have rephrased the text surrounding Eq. (5) to say that at zero detuning it does not converge for $|u| > 1$.

10. Regarding the discussion of the strong coupling regime on page 13, line 410: The optical excitation spectrum was analyzed also at nonzero temperature, both in Ref. 20 and Ref. 21.

We thank the Reviewer for bringing this to our attention.

Action taken: We added a reference and changed the sentence in the revised manuscript (page 14, line 470) to read “In particular, the optical excitation spectrum will be altered by displaying multiple discrete sidebands [20,21].”

11. On page 13, line 415, the possibility of using backaction to manipulate thermal fluctuations beyond a Gaussian distribution is mentioned. Citations would be in order here, as well as reasons for why the creation of non-Gaussian states is a goal in itself.

The sentence referred to by the Reviewer identified the fact that in the regime we describe, dynamical backaction would influence thermal fluctuations in a way that strongly differs from the regular linear case. We are not aware of works that explore the effects and potential use of nonlinear dynamical backaction (at least, away from the single-photon strong coupling regime $g_0 \approx \kappa$). Given the fact that thermal noise plays a crucial role in many applications such as (nano)mechanical sensing, its manipulation is generally a topic of interest. For example, regular dynamical backaction can be used to alter (e.g. enhance) sensor bandwidth. As such, we believe it could be a worthwhile future undertaking to consider how nonlinear backaction, influencing thermal motion beyond Gaussian distributions, could be employed in particular sensing applications and protocols.

Action taken: In the revised manuscript, we rephrased and expanded the text (page 14, line 474-480) to “As the manipulation of mechanical thermal noise is a topic of interest, e.g. in sensing applications, it could be worthwhile to investigate the effects of the combination of dynamical backaction and the optomechanical nonlinearity. Notably, it would allow to manipulate thermal fluctuations beyond a Gaussian distribution.” We did not repeat references that focus on nonlinear backaction in the single-photon strong coupling regime to avoid giving the impression that our present devices could operate there.

Response to Reviewer #3:

We thank the Reviewer for his/her careful reading and valuable comments to our manuscript. We are glad that the Reviewer recognizes our work as the first experiment to explore a qualitatively new regime, and that our demonstrations “will inspire thinking about new nonlinear optomechanical protocols”.

The Reviewer raises some excellent points, especially about the section that describes quadratic measurements. We have taken all suggestions to heart in revising our manuscript. We answer to each of the points step-by-step below, and list the actions we have taken to improve our manuscript.

1) I find the section “Quadratic measurement of motion” to be potentially misleading. Eq. (3) as defined appears to be a correct expression for the peak to shot-noise-background SNR in an integration time of $1/\Gamma$. However, the relevant signal to noise for single-phonon

converges when the absolute value of the second term between square brackets is smaller than unity, which results in the identity we quote.

detection or non-classical conditional state preparation should be taken over an integration time of the quantum state lifetime $\sim 1/(\Gamma n_{\text{th}})$. For the same reason, the factor of n_{th} is (correctly) included in the inequality on page 5, line 431. For MHz scale mechanical frequencies n_{th} is a large factor even at cryogenic temperatures. Perhaps the authors could point out if there are any applications that become possible by satisfying Eq. (3) with $n_{\text{min}}=1$ (that could not be accomplished by just making a linear position measurement and squaring the results).

We thank the Reviewer for this valuable comment. We introduced eq. (3) to give context and a sense of scale for the measurement imprecision in our nonlinear measurements. We agree that in order to use these measurements for quantum state preparation the nonlinear measurement rate should exceed the thermal decoherence rate. This amounts to reaching an imprecision $n_{\text{min}} < (2n_{\text{th}})^{-1/2}$. We realize that in the original manuscript our wording was not chosen carefully enough to describe the meaning of n_{min} .

Action taken: We have significantly rewritten the section on quadratic measurement of motion based on the Reviewer's comments. This includes in particular the introduction and discussion of Eq. 3. To avoid confusion, we renamed the quantity n_{min} to n_{imp} . We de-emphasized the impact of reaching $n_{\text{imp}}=1$, and we added text at the end of the paragraph that discusses the requirements for quantum state preparation explicitly.

The Quadratic measurement section also sweeps under the rug the question of linear backaction, only briefly mentioned on line 273. The criteria for low linear measurement backaction as laid out in Ref [24], (as I understand it, Although I do not understand all of the subtle exceptions described in [24]), is that either $g_0 > \kappa$ or roughly $g_0 > \kappa \sqrt{1-\eta}$ with the addition of active feedback to suppress the backaction. The current work is still far (several orders of magnitude) from satisfying this criteria even if the detection efficiency is improved by amount suggested by the authors. As it is written now, the manuscript (around line 302) may be misread to imply that all that is necessary is to improve the suppression of linear transduction and not the suppression of linear backaction.

As these quantum state measurement and production ideas seem to be the main purpose of quadratic measurements, the authors should more carefully assess the suitability of their system

Indeed, in our original manuscript we had only briefly mentioned the need for active feedback with a reference to [24]. The Reviewer is correct in stating that significant further efforts to increase in measurement sensitivity and optical quality factor are needed to satisfy the condition to suppress backaction down to the quantum level. This condition reads $g_0^2/\kappa^2 > (1 - \zeta)/8\zeta$, where $\zeta = h\eta$, and h is a factor explicitly taking into account also the technical limitations of the measurement (i.e., the quantum efficiency of the homodyne detector) as we elaborate below.

Action taken: In the revised section, we added an explicit discussion regarding the requirements for measurement-based quantum state preparation. We explain the need to suppress linear backaction through active feedback, introduce the above inequality condition, and discuss the need for further system development.

minor concerns:

2) Are there any indications of instabilities in this system, or are the light levels too low to induce self oscillations, bistability, photothermal or other parametric instabilities. What limits the light level to the 100nW scale? Also, are there any other noise sources that contribute to the cavity line broadening, higher order mechanical modes, low frequency acoustic noise, etc.

These are excellent questions. There are multiple factors that limit the power we employed. As we explain and show in Fig. 5, at higher laser powers the effect of the nonlinear regime on the optical spring effect means that effective mechanical linewidths are strongly broadened. Beyond ~ 100 nW, we see that through the same mechanism low-frequency noise of the laser is imprinted on the mechanical motion as a random drifting of the

mechanical frequency. At higher powers still, we observe instabilities resulting in self-oscillation. The nonlinear dynamics with these large fluctuations are however very complex: The thermomechanically-induced cavity frequency modulations are now larger than the amplitude at which a self-oscillating state would normally saturate. We believe this leads to observed irregular behavior, including random hopping of the oscillation frequency between the two mechanical modes, and occasional occurrence of instabilities for both blue and red detuning.

Regarding other noise sources that could contribute to the cavity line broadening: At the employed powers, any such contribution is negligible compared to the thermomechanical fluctuations of the two mechanical modes we describe, and our cavity frequencies are stable over the duration of our measurements.

Action taken: We introduced a discussion regarding the limits of optical power on page 11, line 356, and on page 13, line 428-437, briefly explaining the effects we observe at higher powers. Their detailed investigation would require new and highly nontrivial theoretical modeling that is beyond the scope of the current work.

3) A few more details about the system would be useful. It is unclear what the 1.3% η includes. Is that just the spatial mode matching from the input round Gaussian mode to whatever the cavity emission pattern looks like. Is there internal loss in the cavity, or other losses in the homodyne detection system (photodetector quantum efficiency, additional mode matching losses, etc). All of these factors seem relevant for measurement based state preparation.

We thank the Reviewer for this suggestion of providing more detail on the system and coupling efficiency. The 1.3% efficiency we quote is derived from a measurement of the optical spring effect at low temperature, as explained in section 4 of the SI. This measurement provides an estimate of the intracavity photon number when combined with our knowledge of g_0 and κ . Relating this to the incident power (measured before the objective lens) allows deriving an input coupling efficiency. This number indeed contains the spatial mode matching of the nanocavity to the incident Gaussian beam, as well as losses at the cryostat windows and objective lens. As such, this efficiency also provides a reasonable estimate of outcoupling efficiency, including the losses at windows and objective lens. However, it does not include some of the technical limitations of our homodyne detector (finite overlap with the local oscillator beam, photodiode quantum efficiency (80%) and the losses at the AR-coated optical elements between objective lens and detector and a variable aperture we use to balance powers on the detectors).

These technical limitation would indeed be important for the feedback scheme we described above and we have written the equation for the feedback scheme using a parameter $\zeta = h\eta$ that includes also these technical limitations in the parameter h . We however note that we did not try extensively to optimize them in this manuscript. We estimate that for these measurements $h \approx 25\%$.

Action taken: In the revised manuscript, we have been more explicit about the fact that η stands for the external coupling efficiency of the cavity and that for the feedback scheme the quantum efficiency of the measurement setup should be included. See especially the end of the largely rewritten quadratic measurement section (page 11, lines 368-373). In Supplementary Note 4 (lines 109-118), we also discuss now the technical limitations of our current setup.

We hope that with these answers and changes, as well as those to all other Reviewers, the Reviewer deems our manuscript suitable for publication in Nature Communications.

REVIEWERS' COMMENTS:

Reviewer #1 (Remarks to the Author):

It is my opinion that the authors have fully satisfied all of my, and the other reviewers, concerns. This is an excellent piece of work, and fully deserves to be published in Nature Communications.

Reviewer #2 (Remarks to the Author):

I find that the authors have addressed the points raised by the referees in a thorough and satisfactory manner. I find the paper to be a valuable addition to the literature and I recommend publication.

Reviewer #3 (Remarks to the Author):

All of my previous concerns have been addressed. I recommend publication of this paper.